# Selective Immobilization of His-Tagged Enzyme on Ni-Chelated Ion Exchange Resin and Its Application in Protein Purification

**DOI:** 10.3390/ijms24043864

**Published:** 2023-02-15

**Authors:** Kangjing Wang, Liting Zhao, Ting Li, Qian Wang, Zhongyang Ding, Weifu Dong

**Affiliations:** 1Key Laboratory of Synthetic and Biological Colloids, Ministry of Education, School of Chemical and Material Engineering, Jiangnan University, 1800 Lihu Road, Wuxi 214122, China; 2Key Laboratory of Carbohydrate Chemistry and Biotechnology, Ministry of Education, School of Biotechnology, Jiangnan University, 1800 Lihu Road, Wuxi 214122, China; 3Department of Chemistry and Biochemistry, University of South Carolina, Columbia, SC 29208, USA

**Keywords:** ion exchange resin, His-tag, immobilization, enzyme activity, protein purification, reusability

## Abstract

Ion exchange resins are suitable as carriers for immobilized enzymes because of their stable physicochemical properties, appropriate particle size and pore structure, and lower loss in continuous operation. In this paper, we report the application of the Ni-chelated ion exchange resin in the immobilization of His-tagged enzyme and protein purification. Acrylic weak acid cation exchange resin (D113H) was selected from four cationic macroporous resins that could chelate the transition metal ion Ni. The maximum adsorption capacity of Ni was ~198 mg/g. Phosphomannose isomerase (PMI) can be successfully immobilized on Ni-chelated D113H from crude enzyme solution through chelation of transition metal ions with the His-tag on the enzyme. The maximum amount of immobilized PMI on the resin was ~143 mg/g. Notably, the immobilized enzyme showed excellent reusability and maintained 92% of its initial activity with 10 cycles of catalytic reaction. In addition, PMI was successfully purified using an affinity chromatography column prepared by Ni-chelated D113H, which showed the potential for the immobilization and purification process to be realized in one step.

## 1. Introduction

As natural biocatalysts, enzymes accelerate various chemical reactions in nature mildly and efficiently. However, free enzymes have many application limitations, such as poor environmental stability and difficultly to recycle [1]. Enzyme immobilization technology restricts an enzyme to a specific space by physical or chemical methods, and is one of the effective methods to improve enzyme stability, making recovery and reuse of enzymes possible [2]. In addition, through the design of immobilization methods and carrier materials, enzyme catalytic activity and substrate specificity can be retained to a great extent. In recent years, well-developed immobilized enzymes have been widely used in the food industry [3,4], environmental protection [5], biomedicine [6,7,8], and other fields [9].

Ideal immobilized enzyme carrier materials usually have the following characteristics: (1) low price, easy to obtain; (2) good chemical and mechanical stability; (3) large specific surface area and good pore structure; (4) good affinity for enzymes; (5) surfaces that are easily functionalized and attached to by enzymes; (6) convenient recycling and reuse; (7) environmentally friendly [10]. Various materials have been used as matrix, carriers, or supports for enzyme immobilization, such as biopolymers [11], synthetic polymers [12,13], organic–inorganic composites [14], porous ceramics [15], and nanoparticles [16].

Ion exchange resins with acidic or basic groups possess the advantages of low price, good mechanical strength, and stable physical and chemical properties, and also have broad application prospects as immobilized carriers [17]. Some studies have investigated the immobilization of enzymes such as lipase [18,19], inulinase [20], and tyrosinase [21] using ion exchange resin.

Traditional ion exchange resins include gel-type ion exchange resins, macroporous adsorption resins, cross-linked resins, etc. [22]. According to different preparation methods, gel resins include agarose and dextran styrene divinylbenzene resins; macroporous adsorption resins are obtained by suspension copolymerization after adding inert solvent or porogen into the monomer; cross-linked resins are prepared by chloromethylation and Fourier reaction alkylation, using styrene divinylbenzene copolymer as the skeleton, with the specific surface area porosity of the resin then improved by a cross-linking reaction [23].

Macroporous resin has large pores, large surface area, many active centers, high efficiency, and a fast ion diffusion and ion exchange rate, about ten times faster than traditional gel resin. Macroporous resin has many advantages: resistance to swelling, oxidation, heat and temperature change; not being easy to crack, and easier adsorption and exchange of organic macromolecules, giving it strong pollution resistance and making it easier to regenerate. However, the enzyme immobilized by ion exchange resin adopts ion exchange adsorption, which leads to the lack of a tight bond between the resins and enzymes.

In 1956, Professor He Binglin reported that styrene can be copolymerized to obtain ion exchange resin with a macroporous structure [24], and applied this to the separation and purification of natural products and drugs, such as the purification of streptomycin with the amino group by macroporous ion exchange resin. Ion exchange chromatography uses the force between the charge on the protein and the charged group on the ion exchange filler to complete the adsorption. The difference in the interaction force between these charges leads to different adsorption intensities, which will make different proteins elute successively in the elution process to achieve the separation effect. It is also widely used in the purification of recombinant proteins, antibodies, vaccines, and other substances [25,26]. At present, ion exchange chromatography is the most commonly used method in protein separation and purification [25], and accounting for 75% of total biochemical separation processes.

During the separation and purification of proteins, it was found that the affinity tags such as the imidazolyl group of histidine, the sulfhydryl group of hemilysine, and the indolyl group of tryptophan on the surface of proteins can chelate with transition metal ions (Ni^2+^, Co^2+^, Zn^2+^, Cu^2+^) under certain conditions to form stable complexes. Affinity tags can be introduced by gene recombination technology, usually by fusing the “tag” into the N- or C- of the target enzymes. Among the numerous tagged recombinant proteins, more than 60% are produced by proteins including a hexahistidine tag (His-tagged) [27].

Inspired by these, we studied the immobilization of enzymes on transition metal ions chelated on ion exchange resin used in His-tagged protein purification. Recombinant enzyme phosphomannose isomerase (PMI) was applied as the model enzyme. PMI plays an essential role in connecting glycometabolism and glycosylation. In prokaryotes and eukaryotes, as the first key enzyme for the flow of GDP mannose synthesized in the glycosylation pathway of protein, it then participates in the synthesis of glycoprotein, glycolipids, and the fungal cell wall [28]. In addition, PMI can transform mannose-6-phosphoric acid to fructose-6-phosphoric acid, which enters the glycolysis process. PMI can catalyze the preparation of L-ribose from l-ketose by the enzymatic method and realize production in *Bacillus* [29]. As PMI can transform mannose into a usable state, it is often used as a screening marker for transgenic plants in plant genetic transformation, which is conducive to improving of transformation efficiency [30].

Based on previous work [27], we selected four macroporous ion exchange resins and chelated transition metal ions on the resin by static adsorption. Transition metal ions can easily form combinations with the imidazole residue of His tags. The immobilized PMI showed excellent recyclability in the catalytic reaction. In addition, referring to the ion exchange chromatography, the resin chelated with Ni was installed in the chromatographic column to prepare the resin purification column, and was used in His-tagged protein purification.

## 2. Results 

### 2.1. The Binding Behavior of Ion Exchange Resin with Metal Ions (Ni^2+^, Co^2+^)

Four resins with different pretreatment methods (0.25 g) and deionized water containing Ni^2+^ (100 mL) were added to a stoppered conical flask and shaken in a shaker at 150 r/min and 30 °C for 3 h. After the adsorption reaction, the supernatant was taken and passed through the membrane to measure the Ni concentration. From Figure 1a, the pretreatment method and the resin ionic form had little effect on the adsorption of Ni by the resin, especially D001 and D001H. The resins mentioned below were all pretreated to the ionic type of Na (Figure 1a). 

Four resins with different pretreatment methods (0.25 g) and deionized water containing Ni^2+^ or Co^2+^ (100 mL) were added into stoppered conical flasks and shaken in a shaker at 150 r/min and 30 °C for 3 h. After the adsorption reaction, the supernatant was taken and passed through the membrane to measure the Ni concentration. The adsorption capacity of the resin for Ni^2+^ and Co^2+^ was basically the same, which may be related to the fact that they are all divalent Ⅷ transition metal ions (Figure 1b).

The ion exchange resin was added to deionized water with different Ni concentrations. The adsorption capacity of the resin was detected as a function of the initial Ni concentration and time. The adsorption rate of the resin was speedy, and the maximum adsorption capacity was reached in around 1 h. When the ion concentration of the solution was 500 mg/L, except for D401 (700 mg/L), the adsorption capacity was the largest. In particular, D113H had the largest adsorption capacity, reaching 198.6 mg/g. The adsorption capacity decreased when the ion concentration in the solution continued to increase. This may be because a too-high ion concentration in the solution may hinder chelation between ions and the resin (Figure 1c–f).

### 2.2. Enzyme Immobilization Capacity

Enzyme immobilization capacity refers to the amount of enzyme immobilized on the ion exchange resin (Figure 2). The immobilization capacity (Q, mg/g) for enzymes was calculated by the following equations:(1)$$Q=\frac{\omega_{0}-\omega }{m}V$$
where *ω*_0_ is the initial enzyme concentration (mg/g), *ω* is the enzyme concentration after adsorption (mg/g), *m* is the weight of resins (mg), and *V* is reaction volume (mL).

### 2.3. Enzyme Activity of Immobilized PMI

The enzyme-catalyzed reaction system was adjusted based on Sigdel’s method [31]. The reaction system consisted of 0.5 mM MgCl_2_, 15 mM F6P (substrate), 20 mM phosphate solution (pH 7.5), and ~2.5 μg of enzyme, with a final volume of 500 μL. The reaction mixture was kept at 30 °C for 10 min followed by cooling on ice to slow the reaction,. Then the concentration of F6P and M6P(product) in the reacted solution was detected by HPLC after passing through the membrane. The enzymatic activity of PMI was defined as the amount of F6P converted to M6P per unit time. Here, the relative enzymatic activity of free enzyme was defined as 100% and the relative enzymatic activity of the immobilized enzyme was calculated (Figure 3). 

### 2.4. Reusability and Stability of Immobilized PMI

After the immobilized enzyme catalyzed the enzymatic reaction, it was washed three times with PBS; then, the enzyme-catalyzed reaction was repeated. The activity of the immobilized enzymes did not decrease significantly after catalysis was repeated 10 times, and 92% of the initial enzymatic activity was maintained. The thermostability and pH stability of the immobilized enzyme were basically the same as those of the free enzyme. In particular, after incubation at 50° for two hours, the relative enzymatic activity of the immobilized enzyme was about 3 times that of the pure enzyme (Figure 4).

### 2.5. Purification of PMI from Crude Enzyme Solution by Ni-Chelated Ion Exchange Resin Column

A simple ion exchange resin purification column was made using a chromatography column and Ni-chelated D113H for enzyme purification (Figure 5a). 

## 3. Discussion

Micropores and macropores coexist in macroporous resins (as shown in Figure 6) and easily adsorb organic macromolecules. The pore size and surface area of the resin provide suitable conditions for the combining of enzymes. The pores of macroporous resins are conducive to the diffusion and combination of enzymes, making the enzymes take an appropriate configuration on the resin. Based on this, we chose four macroporous resins, including chelating resin and cationic resins, namely, D401, D001 D001H, and D113H; and their functional groups are listed in Table 1. Cation exchange resin can dissociate cations (such as H or Na) in aqueous solution and adsorb other cations in the solution. Chelating resin can selectively chelate specific metal ions in the form of ionic bonds or coordination bonds from solutions, in which D401 selectively adsorbs divalent metal ions.

Through pretreatment, the soluble impurities of the resin and the ions, solvents, substances not participating in the polymerization reaction and oligomers left by the resin in the manufacturing process can be removed, and the resin can be transformed into the required form. Proper pretreatment of the resins not only improved their stability, but also activated the resins, improved working exchange capacity and effluent quality. Therefore, pretreatment was highly necessary; it also significantly impacted the activity of subsequent immobilized enzymes. The resins mentioned below were all pretreated to the ionic type of Na. For different resin types, more ions were adsorbed by D113H and D401, which may depend on the functional groups of the resin itself. The chelation of transition metal ions by iminodiacetic acid and carboxyl groups was stronger (Figure 1a).

It can be seen from Figure 1b that the amount of enzyme immobilized by Ni-chelated ion exchange resin was higher than that of Co-chelated ion exchange resin. This may be because Ni has a stronger affinity for PMI than Co, allowing it to immobilize more PMI from the crude enzyme solution. Therefore, enzyme immobilization behavior was studied with Ni-chelated ion exchange resin. Compared with the other two resins, D113H and D401 immobilized more PMI (Figure 1c–f). In addition, the theoretical isoelectric point (pI) of PMI is 5.6, which is in the weakly acidic region. The functional group of D113H is a carboxylic acid, which belongs to the weak acidic resin. Furthermore, D113H is more hydrophilic than styrene resin, as it is an acrylic resin. This is more beneficial to enzyme immobilization and activity. Based on these factors, enzyme immobilization behavior was studied using Ni-chelated D113H resin.

A series of PMI solutions of different concentrations ranging from 0.1 to 1.0 mg/mL in PBS buffer were incubated with Ni-chelated D113H resin. With the increased concentration of enzyme solution, the amount of PMI immobilized by the resin increased. When the concentration of enzyme solution was less than 0.5 mg/mL, the amount of immobilized PMI increased very quickly, and it was basically completely adsorbed. As the concentration continued to increase, the PMI resin immobilization rate gradually decreased and reached a plateau; the highest immobilization was 143 mg/g. The variation of the amount of immobilized enzyme with the concentration of the enzyme solution followed the Langmuir model (Figure 2b).

The relevant research on the adsorption of metal ions by the resin in Section 2.1 shows that the maximum amount of Ni that can be absorbed on the resin is 198.6 mg/g. Within this range, the amount of PMI immobilized kept increasing with the increase of Ni content (Figure 2c).

Figure 3 showed that the immobilization did not cause much loss of enzyme activity, and the enzyme activity was basically maintained. This may be the result of many factors. As an acrylic resin, D113H resin has strong hydrophilicity. Furthermore, the immobilization process was achieved by the affinity of Ni adsorbed on the resin with the His-tag of PMI, which minimized the effect on the active center of the enzyme. Meanwhile, the skeleton structure of the resin had a specific protective effect on the enzymes. Macroporous resins have larger pores to facilitate the diffusion and immobilization of enzymes, giving enzymes a proper configuration on the resin. Moreover, immobilization provided a synergy of enriched local enzyme concentrations in the catalytic reaction.

In addition, the resin can immobilize more enzyme without capping with Tris-HCl buffer after adsorption of Ni, but the enzyme activity decreased by 56% compared to that of the free enzymes. The concentration of Tris-HCl buffer had little effect (Figure 3). Although the concentration increased and the amount of immobilized enzyme increased slightly, the enzyme activity basically remained the same. This may be because the resin without buffer treatment still had some functional groups that did not chelate Ni and could react with the impurity proteins in the crude enzyme solution, resulting in non-selective adsorption.

Reusability and stability of immobilized PMI may be related to the structure of the ion exchange resin. The three-dimensional pore structure of macroporous resin not only hinders the contact between the enzyme and the substrate, but also provides specific protection to the enzyme. The enzymes were chelated on the resin by affinity with Ni, and this strong binding force made it difficult for the enzymes to fall off during the catalysis process (Figure 4).

Compared with the enzyme solution purified with Co-NTA gravity affinity chromatography, some heterobands were still seen in the enzyme solution purified with Ni-chelated D113H. In addition to these shallow heterobands, the obvious band at 46.1 kD indicated that the resin purification column can basically achieve the purification of His-tagged PMI from crude enzyme solution (Figure 5). This indicates that Ni-chelated ion exchange resin has the potential to simultaneously immobilize and purify His-tagged enzymes, greatly simplifying the process of enzyme preparation. Moreover, the ion exchange resin has high yield, low price, convenient storage, and simple regeneration, making it of great significance in industrial applications.

## 4. Materials and Methods

### 4.1. Materials

Nickel chloride hexahydrate (NiCl_2_·6H_2_O), cobalt chloride hexahydrate (CoCl_2_·6H_2_O), NaOH, and HCl were purchased from Shanghai Macklin Biochemical Co. (Shanghai, China). Tris-HCl Buffer (1M, pH8.8) was ordered from Aladdin Reagent Co. Macroporous ion exchange resins were obtained from Jiangsu Suqing Water Treatment Engineering Group Co., Ltd. (Jiangyin, China).

Recombinant strain GLpmi-QCDC was obtained from Dr. Zhongyang Ding. Key gene of gl-pmi was cloned from G. lucidum strain CGMCC 5.26 and expressed in *E. coli.* BL21 (DE3). The recombinant strains GLpmi-QCDC was induced and expressed in *E. coli* [32]. LB medium was prepared using the following recipe: tryptone 10.0 (g·L^−1^), yeast extract 5.0 (g·L^−1^), and NaCl 10.0 (g·L^−1^). 1 × PBS buffer included: KH_2_PO_4_ 0.27 (g·L^−1^), Na_2_HPO_4_ 1.42 (g·L^−1^), NaCl 8.00 (g·L^−1^), and KCl 0.20 (g·L^−1^); the pH was adjusted to pH 7.4 using HCl.

### 4.2. Pretreatment of Resin

First, the resin was washed with distilled water 5 times, stirred in HCl (1.0 mol L^−1^) at 100 rpm for 1 h, and washed with distilled water to remove excess HCl. Then, it was stirred in NaOH (1.0 mol L^−1^) at 100 rpm for 1 h and washed with distilled water to remove excess NaOH. Finally, the resin was stirred in HCl (1.0 mol L^−1^) at 100 rpm for 1 h and rinsed with distilled water until the solution was neutral. The ionic form of the obtained resin is H type. In addition, when the order of adding NaOH and HCl is changed during pretreatment, the resulting ionic form is Na type.

### 4.3. Adsorption of Resin with Metal Ions (Ni^2+^ and Co^2+^)

Ni^2+^ aqueous solutions with different concentrations (100, 300, 500, 700, 1000 mg/L) were prepared. Pretreated ion exchange resin was added to the above solutions. Then, the solutions were shaken and adsorbed in a constant temperature shaker at 150 rpm. The supernatant was collected and filtered with filter membrane to measure the ion concentration. The load capacity (mg·g^−1^) of the ion was calculated by the following equations:(2)$$\mathrm{Load}\;\mathrm{capacity}=\frac{C_{0}-C}{m}V$$
where *C*_0_ is the initial ion concentration (mg/g), *C* is the ion concentration after adsorption (mg/g), and m is the weight of resins (mg), *V* is solution volume (mL).

The obtained Ni-chelated ion exchange resins were shaken at low speed in Tris-HCl buffer (pH 8.0) for 1 h and then washed with deionized water until neutral for later use.

The adsorption of Co on ion exchange resins is the same as that of Ni above.

### 4.4. Induced Expression of PMI

The recombinant strains GLpmi-QCDC was inoculated into liquid LB medium containing 30 mg·L^−1^ kanamycin. After overnight culture, 1% inoculum was inoculated into fresh sterile LB medium. IPTG was added when OD_600_ reached 0.6–0.8. The strain was then cultured at 25 °C and 150 r·min^−1^ for 10 h. The cells were centrifuged at 12,000 r·min^−1^ for 5 min, washed twice with 1 × PBS (pH = 7.4) solution, and suspended with 1 × PBS solution; then, they broken by ultrasound in ice water mixture. At 4 °C, the supernatant was centrifuged at 12,000 rpm for 20 min. The crude enzyme solution was used for enzyme activity determination, protein concentration determination, and SDS-PAGE analysis. Purified enzyme (25 mL, 0.3 mg/mL) can be obtained after purification of crude enzyme solution (100 mL, 1.7 mg/mL).

### 4.5. Separation and Purification of PMI

The crude enzyme solution was filtered by microporous membrane (0.45 μm) and purified by Co-NTA gravity affinity chromatography (Equilibrium buffer: 1 × PBS, NaCl 0.5 mol·L^−1^, imidazole 20 mmol·L^−1^, pH 7.4. Elution buffer: 1 × PBS, NaCl 0.5 mol·L^−1^, imidazole 300 mmol·L^−1^, pH 7.4). The purified PMI was obtained by dialysis with 2 L 1 × PBS buffer 3 times.

### 4.6. Assembly of Resin with PMI

Ni/Co-chelated ion exchange resin was dispersed in 10 mL purified enzyme solution to incubate and assemble at 4 °C. After a period of reaction, each sample was taken and washed three times with PBS and stored in PBS for later use. All the supernatant was collected to measure the enzyme concentration after immobilization.

### 4.7. Preparation of Ion Exchange Resin Column

The ion exchange resins described in Section 2.3 were loaded into a chromatographic column with sieve plates, rinsed with 5 to 10 times the column volume of deionized water at a flow rate of 5 mL/min, and then stored in 20% ethanol solution. The resin column should be equilibrated with 5~10 column volumes of equilibration solution before use, and the flow rate should be controlled to 1 mL/min (equilibration buffer: 1 × PBS, sodium chloride 0.5 mol·L^−1^, imidazole 20 mmol·L^−1^, pH 7.4).

### 4.8. Characterization

The internal structure of the resin was characterized by scanning electron microscopy (SEM Hitachi S-4800). The concentrations of ions were detected using an atomic absorption spectrophotometer (AAS, A3F). Protein concentrations were determined with a microplate reader at a wavelength of 562 nm based on bicinchoninic acid (BCA) kits; the working range of the BCA method was 0.05~2 mg/mL. The concentrations of F6P and M6P were detected by high performance liquid chromatography (HPLC, ICS-5000, Thermo Nicolet Corp., Waltham, MA, USA). The purity and molecular weight of the protein were detected by sodium dodecyl sulfate polyacrylamide gel electrophoresis (SDS-PAGE, Bio-Rad, Hercules, CA, USA). The protein zones were examined by standard Coomassie brilliant blue staining methods.

## 5. Conclusions

In summary, ion exchange resin was modified to obtain Ni-chelated acrylic weak acid cation exchange resin, which was used for immobilization and purification of the His-tagged enzyme. The maximum Ni loaded on the resin was 198 mg/g, and the optimal binding PMI immobilized by the affinity of Ni and His-tag was 143 mg/g. After immobilization, the enzyme activity was basically the same. In relation to reusability, the immobilized PMI maintained about 92% of its initial activity for 10 cycles. An affinity chromatography column can be prepared using Ni-chelated D113H resin, and purified enzyme solution can be obtained by simple filling and elution. Overall, this work reveals that the Ni-chelated ion exchange resin has wide prospects as a superior support for enzyme immobilization and purification applications.

## Figures and Tables

**Figure 1 ijms-24-03864-f001:**
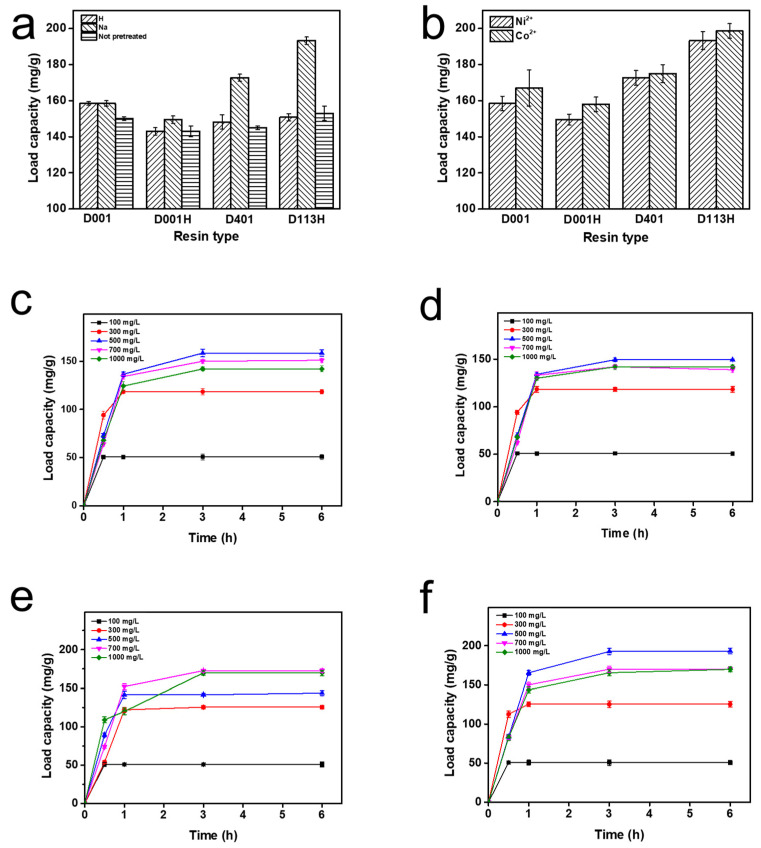
(**a**) Load capacity (mg/g) of resins with different pretreatment methods; (**b**) load capacity (mg/g) of resins with different ions; load capacity (mg/g) of (**c**) D001, (**d**) D001H, (**e**) D401, and (**f**) D113H at different times and ion concentrations.

**Figure 2 ijms-24-03864-f002:**
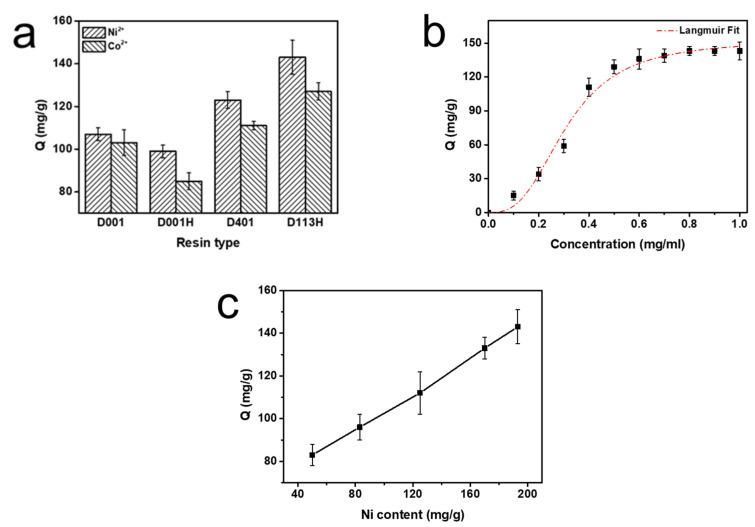
(**a**) Enzyme immobilization capacity (Q, mg/g) of PMI with different resin types, (**b**) enzyme immobilization capacity (mg/g) of PMI on Ni-chelated ion exchange resin at different PMI concentrations, and (**c**) enzyme immobilization capacity (mg/g) of PMI on Ni-chelated ion exchange resin with different Ni-chelating amounts.

**Figure 3 ijms-24-03864-f003:**
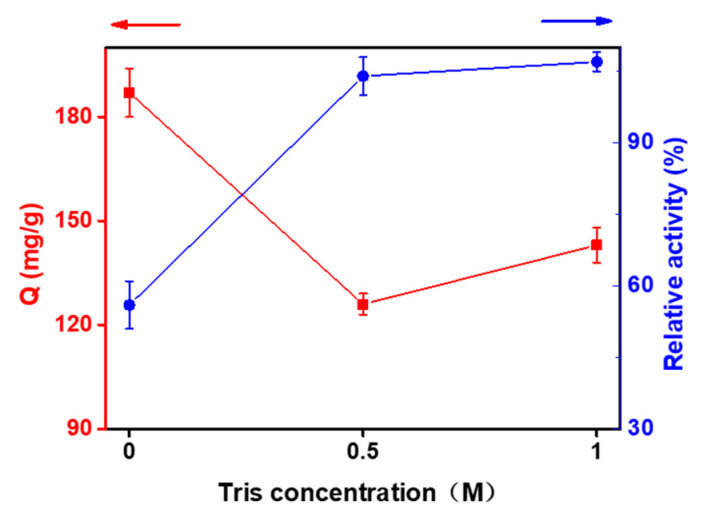
Effect of Tris-HCl on the capacity and activity of immobilized enzyme.

**Figure 4 ijms-24-03864-f004:**
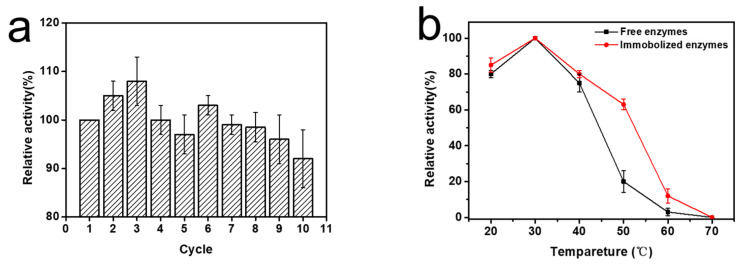
(**a**) Reusability of F6P reaction catalyzed by immobilized enzymes, (**b**) thermostability of immobilized enzymes and free enzymes incubated 2 h at different temperature, and (**c**) pH stability of immobilized enzymes and free enzymes incubated 2 h at different pH.

**Figure 5 ijms-24-03864-f005:**
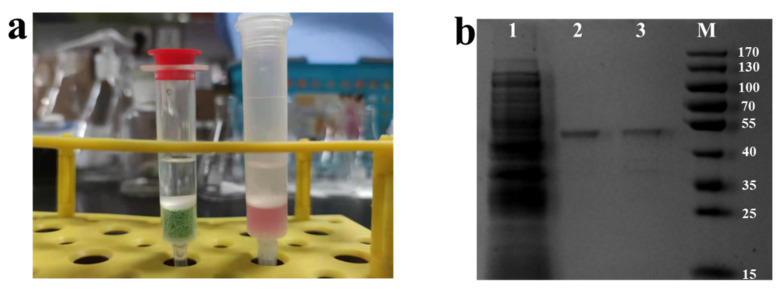
(**a**) Digital photograph of Ni-chelated ion exchange resin column (left) and Co-NTA gravity affinity chromatography (right). (**b**) SDS-PAGE analysis of PMI immobilization from crude enzyme solution: Lane 1—crude enzyme solution (cell disruption supernatant of GLpmi-DCQC); Lane 2—purified enzyme solution by Co-NTA gravity affinity chromatography; Lane 3—purified enzyme solution by Ni-chelated ion exchange resin column after concentration. M: standard protein maker.

**Figure 6 ijms-24-03864-f006:**
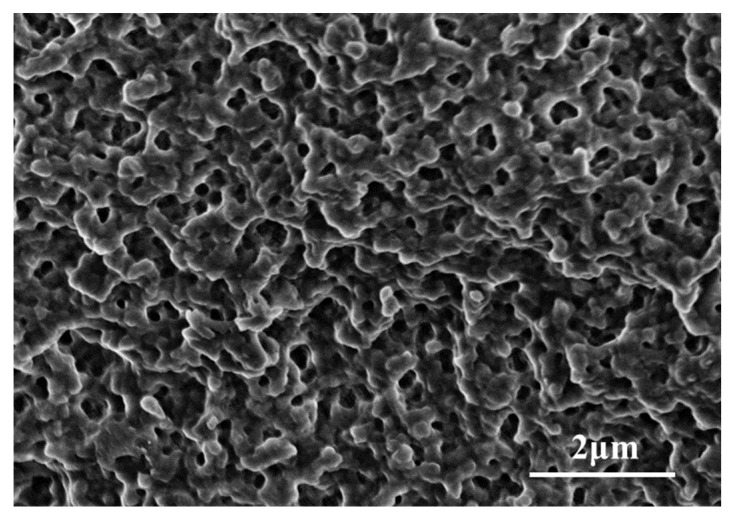
SEM images of the internal structure of ion exchange resin.

**Table 1 ijms-24-03864-t001:** Characteristics of resins.

Resin Type	Functional Group	Ionic Form	Moisture (%)	Particle Size (%)
D401	Iminodiacetic acid	Na	52–58	0.315–1.25 mm
D001	Sulfonic acid group	Na	45–50	0.315–1.25 mm ≥ 95
D001H	Sulfonic acid group	H	48–58	0.315–1.25 mm ≥ 95
D113H	Carboxylic acid	H	45–52	0.315–1.25 mm ≥ 95

## Data Availability

The original data in this study are available from the corresponding authors.

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
