# Peer review of "Selective Immobilization of His-Tagged Enzyme on Ni-Chelated Ion Exchange Resin and Its Application in Protein Purification"

_ijms, 2023, doi:10.3390/ijms24043864_

Round 1

Reviewer 1 Report

The article «Selective immobilization of His-tagged enzyme on Ni chelation ion exchange resin and its application in protein purification» is attracted to a relevant topic and has a high applied value.

The authors present application of the Ni chelated ion exchange resin in immobilization of his-tagged enzyme and protein purification. Phosphomannose isomerase (PMI) was successfully immobilized on Ni chelated acrylic weak acid cation exchange resin D113H from crude enzyme solution through chelation of transition metal ions with his-tag on the enzyme. The immobilized enzyme showed excellent reusability and could still maintain 92% of its initial activity with 10 circles of catalytic reaction. The authors proved potential of D113H for immobilization and purification process in one step.

The article is well structured, written in sufficient detail and logically. The authors conducted an extensive experiment. The introduction is very well written, it will fit the reader to the main idea of the article.

Minor remark:

1. Introduction, second paragraph: 1) Low price … 2) Good chemical … 3) Large specific … 4) Good affinity …etc. should be lowercase.

2. Figures 2, 3, 5 are best done on the whole page. Figure 4, on the contrary, should be made smaller to be comparable with Figures 2a and 3a.

3. Figures 2a and 2b. It is advisable to do it on the same scale, and the y-axis should not start from zero, but from 100-120 and up to 200. This manipulation will make the differences more pronounced.

3. Figure 3a. y-axis should not start from zero, but from 70-80 and up to 160.

4. Section 3.4. This may be the result of many reasons: As an acrylic resin, D113H resin showed good hydrophilicity. As an acrylic … should be lowercase.

5. Figure 5a. y-axis should not start from zero, but from 80 and up to 120.

6. Reference 22. THEORY FOR ELECTROSTATIC INTERACTION CHRO-MATOGRAPHY OF PROTEINS capitals should be removed.

7. Reference 31 line breaks should be removed.

8. All Latin names such as Aspergillus nidulans, Ganoderma lucidum, Bacillus amyloliquefaciens etc. should be italicized.

9. Line numbers should be added.

Author Response

We would like to acknowledge the Reviewer for the valuable comments which, in our opinion, strongly enhanced the quality of this manuscript. We have studied the comments carefully and made corrections accordingly. Here, we provided our responses to the comments point-by-point including the changes with respect to the original version.

Comments and Suggestions for Authors

The article «Selective immobilization of His-tagged enzyme on Ni chelation ion exchange resin and its application in protein purification» is attracted to a relevant topic and has a high applied value.

The authors present application of the Ni chelated ion exchange resin in immobilization of his-tagged enzyme and protein purification. Phosphomannose isomerase (PMI) was successfully immobilized on Ni chelated acrylic weak acid cation exchange resin D113H from crude enzyme solution through chelation of transition metal ions with his-tag on the enzyme. The immobilized enzyme showed excellent reusability and could still maintain 92% of its initial activity with 10 circles of catalytic reaction. The authors proved potential of D113H for immobilization and purification process in one step.

The article is well structured, written in sufficient detail and logically. The authors conducted an extensive experiment. The introduction is very well written, it will fit the reader to the main idea of the article.

Minor remark:

  1. Introduction, second paragraph: 1) Low price … 2) Good chemical … 3) Large specific … 4) Good affinity …etc. should be lowercase.

Answer: Lowercase has been corrected.

  1. Figures 2, 3, 5 are best done on the whole page. Figure 4, on the contrary, should be made smaller to be comparable with Figures 2a and 3a.

Answer: These figures have been revised.

  1. Figures 2a and 2b. It is advisable to do it on the same scale, and the y-axis should not start from zero, but from 100-120 and up to 200. This manipulation will make the differences more pronounced.

Answer: These figures have been revised.

  1. Figure 3a. y-axis should not start from zero, but from 70-80 and up to 160.

Answer: The y-axis has been revised to start from 70.

  1. Section 3.4. This may be the result of many reasons: As an acrylic resin, D113H resin showed good hydrophilicity. As an acrylic … should be lowercase.

Answer: Lowercase has been corrected.

  1. Figure 5a. y-axis should not start from zero, but from 80 and up to 120.

Answer: The y-axis has been resived to start from 80.

  1. Reference 22. THEORY FOR ELECTROSTATIC INTERACTION CHRO-MATOGRAPHY OF PROTEINS capitals should be removed.

Answer: The reference has been revised.

  1. Stahlberg, J.; Jonsson, B.; Horvath, C., Theory for electrostatic interaction chromatography of proteins. Analytical Chemistry 1991, 63 (17), 1867-1874.
  2. Reference 31 line breaks should be removed.

Answer: The reference has been revised.

  1. Li Y., L. Z., Gu Zh., Li Y., Shi G., Ding Zh., Heterologous expression and characterization of the key enzymes involved in sugar donor synthesis of polysaccharide in Ganoderma lucidum. Microbiology China 2019, 46 (12), 3233-3247.
  2. All Latin names such as Aspergillus nidulans, Ganoderma lucidum, Bacillus amyloliquefaciens etc. should be italicized.

Answer: All Latin names have been italicized.

  1. Line numbers should be added.

Answer: Line numbers have been added.

Reviewer 2 Report

Title: Selective immobilization of His-tagged enzyme on Ni chelation  ion exchange resin and its application in protein purification

 The manuscript reports the use of ion exchange resins for purification and immobilization of his-tagged proteins, using a phosphomannose isomerase as a study model. In my opinion, the subject of the manuscript does not bring novelty interesting enough to be published by Int. J. Mol. Sci. and there are many problems concerning the scientific and technical aspects of the manuscript. For instance, the authors claim  (lines 339 to 341, “this work reveals that the Ni chelated ion exchange resin has wide prospects to serve as a  superior support for enzymes immobilization and purification applications”). What is supposed to mean? The authors did not test with at least one commercial resin; no work was discussed to have some parameters for comparison!

 Furthermore, it is hard to understand the real significance of what the authors have done and the interpretation of the results. For instance, item 3.3 (Enzyme immobilization capacity), sorry, but  I needed to re-read the manuscript a few times to understand what was done partially. There are no data reported about the enzyme activity (U mg or U mL) after purification and before purification. Thus, there is no data about the efficiency of the process! Furthermore,  the use of the term “immobilization capacityis absolutely equivocated! By the equation, the authors mean recovered activity, I suppose. Another important point: why was used "g of enzyme" per g of support (I suppose) and not units of activity? The information given here is INSUFFICIENT for the reader to know exactly what you did. This is a problem, because your results will depend on EXACTLY how you did this experiment.

 In the line 253 : “The variation of the amount of immobilized enzyme with the  concentration of the enzyme solution followed the Langmuir model.” What is supposed to mean? Is Langmuir adsorption model? How did you calculate Langmuir adsorption isotherm parameters? And about the adsorption experiments? Unfortunately, the materials & methods section does not describe in detail how the experiments were done. Indeed, I would say that, in some cases, it does not describe the procedure at all.

In the item 3.4: Enzyme activity of immobilized PMI

Line 267 – 271 : How, EXACTLY, was this assay done? Again, unfortunately, the materials and methods section does not make this sufficiently clear (it essentially does not give any more information than that which you give here). If the materials and methods section does not make this sufficiently clear (it practically does not give complete information), the results' interpretation is impaired.  

Line 281: “Moreover, immobilization provided a synergy of enriched local enzyme concentrations and better synergistic effect in the catalytic reactionsorry, but the reader did not understand what was done and what experimental data were used to the authors claim this.

Thus, it is not very easy to understand and validate the data presented in the manuscript.

Due to these problems, I do not recommend the publication of this manuscript.

Author Response

We would like to acknowledge the Reviewer for the valuable comments which, in our opinion, strongly enhanced the quality of this manuscript. We have studied the comments carefully and made corrections accordingly. Here, we provided our responses to the comments point-by-point including the changes with respect to the original version.

The quality and clarity of the text (grammar, spelling, punctuation etc.) have been carefully checked.

Title: Selective immobilization of His-tagged enzyme on Ni chelation ion exchange resin and its application in protein purification

 The manuscript reports the use of ion exchange resins for purification and immobilization of his-tagged proteins, using a phosphomannose isomerase as a study model. In my opinion, the subject of the manuscript does not bring novelty interesting enough to be published by Int. J. Mol. Sci. and there are many problems concerning the scientific and technical aspects of the manuscript.

  1. For instance, the authors claim (lines 339 to 341, “this work reveals that the Ni chelated ion exchange resin has wide prospects to serve as a superior support for enzymes immobilization and purification applications”). What is supposed to mean? The authors did not test with at least one commercial resin; no work was discussed to have some parameters for comparison!

Answer: In this work, four commercial resins were selected, including D401(Suqing selective and chelating ion exchange resin), D001, D001H(Suqing poly(St-DVB)based microporous type strong acidic cation exchange resins) and D113H(Suqing polyacrylate based microporous type weak acidic cation exchange resin). Fig.4(relative activity expressed by blue line) showed that the activity of the enzyme was basically maintained after immobilization. Furthermore, the purification effects of prepared ion exchange resin purification column and commercial Co-NTA gravity affinity chromatography were compared in item 3.6. The obvious band of target enzyme was obtained successfully. Based on the above, it is considered that the Ni chelated ion exchange resin can be used for enzymes immobilization and purification applications.

  1. Furthermore, it is hard to understand the real significance of what the authors have done and the interpretation of the results. For instance, item 3.3 (Enzyme immobilization capacity), sorry, but I needed to re-read the manuscript a few times to understand what was done partially. There are no data reported about the enzyme activity (U mg or U mL) after purification and before purification. Thus, there is no data about the efficiency of the process! Furthermore, the use of the term “immobilization capacity” is absolutely equivocated! By the equation, the authors mean recovered activity, I suppose. Another important point: why was used "g of enzyme" per g of support (I suppose) and not units of activity? The information given here is INSUFFICIENT for the reader to know exactly what you did. This is a problem, because your results will depend on EXACTLY how you did this experiment.

Answer: The definition of immobilization capacity has been added in Page 7 Line 232: Enzyme immobilization capacity refers to the amount of enzymes immobilized on the ion exchange resin. The enzyme activity was described in item 3.4: the enzymatic activity of PMI(U/mg) was defined as the amount of F6P(substrate) converted to M6P(product) per unit time. Here, the relative enzymatic activity of free enzyme was defined as 100%, and the relative enzymatic activity of immobilized enzyme was calculated.

  1. In the line 253: “The variation of the amount of immobilized enzyme with the concentration of the enzyme solution followed the Langmuir model.” What is supposed to mean? Is Langmuir adsorption model? How did you calculate Langmuir adsorption isotherm parameters? And about the adsorption experiments? Unfortunately, the materials & methodssection does not describe in detail how the experiments were done. Indeed, I would say that, in some cases, it does not describe the procedure at all.

Answer: The experimental has been added in Line 249: A series of PMI solutions of different concentrations ranging from 0.1 to 1.0 mg/mL in PBS buffer were incubated with Ni chelated D113H resin. It can be clearly observed that the immobilization amount of Ni chelated D113H resin toward PMI increased rapidly with the increasing initial concentration of MPI and reached maximum at the concentration of 0.8 mg/mL. After fitting the obtained immobilization amount, it is found that Langmuir model was suitable for describing the kinetic data.

  1. In the item 3.4: Enzyme activity of immobilized PMI. Line 267 – 271: How, EXACTLY, was this assay done? Again, unfortunately, the materials and methods section does not make this sufficiently clear (it essentially does not give any more information than that which you give here). If the materials and methods section does not make this sufficiently clear (it practically does not give complete information), the results' interpretation is impaired.  

Answer: The assay has been supplemented in Line 269: The reaction system consisted of 0.5 mM MgCl2, 15 mM F6P (substrate), 20 mM phosphate solution (pH 7.5), ~2.5 μg of enzyme in a final volume of 500 μl. The reaction mixture was kept at 30°C for 10 min followed by cooling on ice to slow the reaction. Then the concentration of F6P and M6P(product) in the reacted solution was detected by HPLC after passing through the membrane.

  1. Line 281: “Moreover, immobilization provided a synergy of enriched local enzyme concentrations and better synergistic effect in the catalytic reaction” sorry, but the reader did not understand what was done and what experimental data were used tothe authors claim this.

 Answer: In the analysis of the results, it was concerned that after enzyme immobilization, there will be synergistic effects due to enriched local enzyme concentrations. After thinking, it is really inappropriate. Thanks for pointing out this, which has been revised in Line 283: Moreover, immobilization provided a synergy of enriched local enzyme concentrations in the catalytic reaction.

Round 2

Reviewer 2 Report

We would like to acknowledge the Reviewer for the valuable comments which, in our opinion, strongly enhanced the quality of this manuscript. We have studied the comments carefully and made corrections accordingly. Here, we provided our responses to the comments point-by-point including the changes with respect to the original version.

The quality and clarity of the text (grammar, spelling, punctuation etc.) have been carefully checked.

Reply: The manuscript still has several grammatical errors and a lack of standard units ... beyond  basic conceptual errors in biochemistry, for example, in line 113: The recombinant strains GLpmi-QCDC was induced and expressed in E. coli.

Title: Selective immobilization of His-tagged enzyme on Ni chelation ion exchange resin and its application in protein purification

The manuscript reports the use of ion exchange resins for purification and immobilization of his-tagged proteins, using a phosphomannose isomerase as a study model. In my opinion, the subject of the manuscript does not bring novelty interesting enough to be published by Int. J. Mol. Sci. and there are many problems concerning the scientific and technical aspects of the manuscript.

  1. For instance, the authors claim (lines 339 to 341, “this work reveals that the Ni chelated ion exchange resin has wide prospects to serve as a superior support for enzymes immobilization and purification applications”). What is supposed to mean? The authors did not test with at least one commercial resin; no work was discussed to have some parameters for comparison!

Answer: In this work, four commercial resins were selected, including D401(Suqing selective and chelating ion exchange resin), D001, D001H(Suqing poly(St-DVB)based microporous type strong acidic cation exchange resins) and D113H(Suqing polyacrylate based microporous type weak acidic cation exchange resin). Fig.4(relative activity expressed by blue line) showed that the activity of the enzyme was basically maintained after immobilization. Furthermore, the purification effects of prepared ion exchange resin purification column and commercial Co-NTA gravity affinity chromatography were compared in item 3.6. The obvious band of target enzyme was obtained successfully. Based on the above, it is considered that the Ni chelated ion exchange resin can be used for enzymes immobilization and purification applications.

Reply: i) the authors did not answer my question. The authors did not evaluate any commercially available resin for purification of his-tag proteins, such as a simple and classic resin such as Ni Sepharose, for example, which has been available for more than 2 decades.

ii) “In this work, four commercial resins were selected, including D401(Suqing selective and chelating ion exchange resin), D001, D001H (Suqing poly(St-DVB)based microporous type strong acidic cation exchange resins) and D113H(Suqing polyacrylate based microporous type weak acidic cation exchange resin)” why this type of information was not added in the manuscript? why is this not described anywhere? At some points of the manuscript, the authors write about the composition, for example: “line 278: as an acrylic resin, D113H resin showed good hydrophilicity”. The reader has no information on the composition of the resin! the authors do not provide enough information about this in M&M. It is very complicated to assume that the reader must search for basic information in other references to try to understand what was done!

iii) Fig.4(relative activity expressed by blue line) showed that the activity of the enzyme was basically maintained after immobilization”. Sorry, but Fig.4 showed (line 295) Effect of Tris-HCl on the capacity and activity of immobilized enzyme. It does not appear to show anything about the activity of the immobilized enzyme. What is supposed to mean  “ the Effect of Tris-HCl”  There is no information about this essay in M&M!

  1. Furthermore, it is hard to understand the real significance of what the authors have done and the interpretation of the results. For instance, item 3.3 (Enzyme immobilization capacity), sorry, but I needed to re-read the manuscript a few times to understand what was done partially. There are no data reported about the enzyme activity (U mg or U mL) after purification and before purification. Thus, there is no data about the efficiency of the process! Furthermore, the use of the term “immobilization capacity” is absolutely equivocated! By the equation, the authors mean recovered activity, I suppose. Another important point: why was used "g of enzyme" per g of support (I suppose) and not units of activity? The information given here is INSUFFICIENT for the reader to know exactly what you did. This is a problem, because your results will depend on EXACTLY how you did this experiment.

Answer: The definition of immobilization capacity has been added in Page 7 Line 232: Enzyme immobilization capacity refers to the amount of enzymes immobilized on the ion exchange resin. The enzyme activity was described in item 3.4: the enzymatic activity of PMI(U/mg) was defined as the amount of F6P(substrate) converted to M6P(product) per unit time. Here, the relative enzymatic activity of free enzyme was defined as 100%, and the relative enzymatic activity of immobilized enzyme was calculated.

Reply: Again, the authors did not answer my question. No activity data was reported! why? How was the activity of the immobilized enzyme calculated? Moreover, the definition of activity is wrong! please see the IUBMB.

In short, the manuscript was not improved and still is very badly written. Note that I could have done many more comments. Because its subject is poor, I have no reason to continue revising the manuscript point-by-point.